# Interference control in working memory: Evidence for discriminant validity between removal and inhibition tasks

**Alodie Rey-Mermet**[1,2‡]*, **Krishneil A. Singh**[3‡], **Gilles E. Gignac**[3], **Christopher R. Brydges**[3,4], **Ullrich K. H. Ecker**[3]

**1** Faculty of Psychology, Swiss Distance University Institute, Brig, Switzerland, **2** Department of Psychology, Cognitive Psychology Unit, University of Zurich, Zurich, Switzerland, **3** School of Psychological Science, University of Western Australia, Perth, Australia, **4** West Coast Metabolomics Center, University of California, Davis, United States of America

‡ These authors share first authorship on this work.
* alodie.rey-mermet@fernuni.ch

**Data Availability Statement:** The data, scripts analyses and results presented in this work can be found at the Open Science Framework database (osf.io/c8qb2). A preprint was uploaded at PsyArXiv (doi: 10.31234/osf.io/hdks9).

## Abstract

Working memory (WM) is a system for maintenance of and access to a limited number of goal-relevant representations in the service of higher cognition. Because of its limited capacity, WM requires interference-control processes, allowing us to avoid being distracted by irrelevant information. Recent research has proposed two interference-control processes, which are conceptually similar: (1) an active, item-wise removal process assumed to remove no-longer relevant information from WM, and (2) an inhibitory process assumed to suppress the activation of distractors against competing, goal-relevant representations. The purpose of this study was to determine the extent to which the tasks used to assess removal and inhibition measure the same interference-control construct. Results showed acceptable to good reliabilities for nearly all measures. Similar to previous studies, a structural equation modeling approach identified a reliable latent variable of removal. However, also similar to some previous studies, no latent variable of inhibition could be established. This was the case even when the correlation matrix used to compute the latent variable of inhibition was disattenuated for imperfect reliability. Critically, the individual measures of inhibition were unrelated to the latent variable of removal. These results provide tentative support for the notion that removal is not related to the interference-control processes assessed in inhibition tasks. This suggests that the removal process should be conceptualized as a process independent of the concept of inhibition, as proposed in computational WM models that implement removal as the "unbinding" of a WM item from the context in which it occurred.

## Introduction

Working memory (WM) refers to the ability to maintain and manipulate a limited amount of information in the service of higher cognition [e.g., 1, 2]. WM representations are required to be both stable and flexible. Stability is achieved by maintenance processes that protect relevant

**Funding:** This work was financially supported by a University of Western Australia Postgraduate Award and Safety-Net Top-Up scholarship awarded to KS. The funders had no role in study design, data collection and analysis, decision to publish, or preparation of the manuscript.

**Competing interests:** The authors have declared that no competing interests exist.

WM representations from forgetting and interference from outdated/irrelevant representations. Flexibility is achieved by updating processes that replace outdated/irrelevant representations with goal-relevant representations based on changes in the environment. It has been argued that both the maintenance and updating of information in WM is facilitated by an active, item-wise removal process, which (1) minimizes interference from distractors and (2) frees up capacity for encoding of relevant information [3–9]. Conceptually, this removal process appears to be a similar cognitive process to inhibition, a construct that refers to the ability to ignore and/or suppress distracting information [e.g., 10, 11]. Thus, the tasks used to assess removal and inhibition have been identified as measuring similar interference-control functions in WM. The purpose of the present study was to examine whether these tasks measure the same cognitive process.

## Removal

Recent WM research has proposed that active, item-wise removal is a key process underlying performance in WM tasks [e.g., 3–9]. To specifically measure removal devoid of more generic WM maintenance and retrieval processing, Ecker and colleagues [3, 5, 9] modified an updating task designed by Kessler and Meiran [12]. In this task, participants encode a set of three items (presented in individual frames) before repeatedly replacing individual outdated items with newer items prior to a final recall of the currently memorized set. The task has three critical features: First, on each updating step, participants are instructed to press a key to indicate when they have successfully updated their WM set. This key press provides a measure of updating response time (RT). Second, the to-be-updated item is pre-cued (i.e., the frame of the to-be-updated item turns bold and red) so that participants know what item is being replaced before they are presented with the new item. Third, the time between the onset of this cue and the presentation of the new item (i.e., the cue-target interval, CTI) is either short (e.g., 200 ms) or long (e.g., 1500 ms). This is based on the idea that people can potentially use the CTI to actively remove items from their WM. More specifically, a short CTI would only allow participants to focus their attention on the to-be-updated frame without being able to initiate removal, while a long CTI would allow participants to remove the outdated item representation from WM. Accordingly, in these studies [3, 5, 9], RTs were found to be relatively long in the short-CTI condition (where RTs would include both the time taken to remove the outdated item and the time taken to encode the new item) and relatively short in the long-CTI condition (where RTs would mainly include the time taken to encode the new item).

The validity of removal as a psychometric construct was further established in a recent set of individual-differences studies using structural equation modelling (SEM). Singh and colleagues [9] investigated the relations between removal ability, WM capacity, and fluid intelligence. In that study, removal was significantly associated with both WM capacity and fluid intelligence, however, the relationship between removal and fluid intelligence was entirely mediated by WM capacity. This suggests that removing irrelevant information from WM may contribute to performance on fluid intelligence tasks by effectively increasing operational WM capacity. In summary, numerous studies have proposed the notion of an active, item-wise removal process as an interference-control mechanism in the service of both WM maintenance and WM updating.

## Inhibition

Inhibition is often conceptualized as a general ability underlying attentional control or executive functions, viz. a set of executive abilities allowing us to supervise and control thoughts and actions in order to achieve current goals. Miyake and colleagues [13] reported a model

whereby attentional control was represented by three separate functions: inhibition, updating, and task-switching (i.e., the ability to shift attention to other tasks or perceptual dimensions) [but see 14, 15]. In more recent work, Miyake and Friedman [10, 16] have posited that the tasks used to measure inhibition involve a general ability that underlies all three executive functions [see also 17]. In line with this assumption, research on attentional control has tended to use tasks measuring inhibition [e.g., 18–26]. The results of these studies suggest that attentional control relates to both WM and fluid intelligence [but see 27–31, 32, for some exceptions]. As such, inhibition is widely regarded as a primary construct that is fundamental to higher cognition.

It should be noted, though, that recent research has revealed some difficulties in establishing attentional control—and in particular, inhibition—as a latent variable. Some studies have pointed out low reliability estimates for inhibition measures, possibly explaining the observed low zero-order correlations between these measures [e.g., 33–35]. Furthermore, the inhibition factor has been characterized by large residual variances (i.e., larger than .90) [e.g., 11, 20, 27, 29, 31], or has been dominated by one high-loading (~ .70) measure with low factor loadings for other measures (i.e., < .40) [see, e.g., 11, 20, 23, 24, 31, 36]. This questions the extent to which the inhibition factor is a coherent latent variable representing substantial common variance across different measures. Overall, this research has led some authors to the conclusion that the tasks used to assess inhibition only assess the ability to reduce the interference arising in that particular task [11, 32]. In this case, the inhibition tasks mostly tap into interference-control processes, which are highly task specific.

In summary, previous individual-differences research has put forward inhibition as a core interference-control mechanism related to WM. Nevertheless, there is some evidence that establishing inhibition as a valid and reliable construct at the latent-variable level is more difficult than previously thought. This emphasizes that inhibition might not be a general construct, thus mandating us to be very cautious when referring to the concept of "inhibition".

## The present study

The purpose of the present study was to investigate the extent to which the tasks used to assess removal and inhibition measure the same construct. In addition to the conceptual similarity between removal and inhibition abilities, there is empirical support for the assumption that the tasks used to assess removal and inhibition measure similar constructs. First, the removal process has been argued to be involved in WM and updating task performance [e.g., 3–9]. This implies that correlations between (1) WM and attentional control, and between (2) WM updating and inhibition may be driven by a common removal/inhibition factor. Second, individual differences in both removal efficiency and inhibition have been found to be related to individual differences in WM capacity and fluid intelligence [9, 18–26, 30, 37].

In the present study, we used a confirmatory factor analysis (CFA) approach to account for the correlational structure among the measures of removal and inhibition. Removal was measured using the updating task battery of Ecker and colleagues [3, 5, 9], with letters, digits, and words as stimuli. Inhibition was measured using tasks that are broadly (though not universally) assumed to require inhibition and were used in previous individual-differences research [e.g., 11, 13, 32, 38]. Given that, as reviewed above, previous investigations have reported difficulties establishing inhibition at the latent-variable level [see 39, for an overview] and measures of inhibition are known to be associated with imperfect reliability [e.g., 20, 27, 40, 41], we also conducted a CFA on a correlation matrix disattenuated for imperfect reliability in order to increase the chances of observing a coherent latent variable of inhibition [42].

It was hypothesized that if the tasks used to assess removal and inhibition measure the same cognitive process, zero-order correlations should be high between all measures. Moreover, either all measures should load on a single factor, or, if separate factors for removal and inhibition were to emerge, these two factors should be highly correlated. In contrast, if the tasks used to assess removal and inhibition measure different cognitive processes, zero-order correlations should be low between the removal measures and the inhibition measures. There should also be no significant correlation between removal and inhibition factors. Moreover, in case there is no general construct of inhibition and thus no coherent factor of inhibition were to emerge, as suggested by recent research [e.g., 11, 14, 32], we planned to investigate the relations between each individual measure of inhibition and the latent variable of removal to determine.

## Method

### Participants

In total, 138 undergraduate students from the University of Western Australia participated in the study for partial course credit. Three participants were excluded because they did not complete the inhibition tasks. Sixteen participants were excluded because their performance in any of the removal or inhibition tasks fell 3 *SD* below the respective mean of all participants. Eight participants were removed because they had a low number of trials (fewer than 40 trials) in any task. In addition, the data of one participant were excluded as they were identified as a multivariate outlier (i.e., significant Mahalanobis's $d^2$ values) when checking for multivariate normality using Mardia's [43] kurtosis index. The final sample consisted of 110 participants (66 females; age: *M* = 20.46 years, *SD* = 3.08; age range: 18 to 36 years).

We estimated the recommended sample size for a measurement model including two latent variables (i.e., removal and inhibition) and nine manifest variables (i.e., three removal measures and five inhibition measures). To this end, we used a sample size calculator for SEM [44, 45] with a statistical power of 0.80 and a probability level of 0.05. As previous research focused on two correlations–that is, the correlation between WM and attentional control, and the correlation between WM updating and inhibition–we estimated the sample size using two anticipated effect sizes. That is, we computed a first anticipated effect size as the average of the latent correlations between attentional control and WM (i.e., .29 when calculated with the studies listed in the first table of Rey-Mermet and colleagues [32]). In this case, the minimum sample size should be 97 participants. We computed a second anticipated effect size as the average of the latent correlations between inhibition and updating (i.e., .59 when computed with [13, 19, 28, 46–48]). In this case, the minimum sample size should be 88 participants. Together, this suggests that in addition to the general advantages of using confirmatory factor analysis even when the sample size may not be considered as particularly large (i.e., around 100 [see 49, 50]), our sample size of 110 participants should be sufficient for the CFA.

### Materials

**Updating tasks.** Removal efficiency was measured using three updating tasks based on the paradigm introduced by Ecker and colleagues [3, 5, 9]. The three tasks only differed with respect to the stimulus materials (i.e., letters, digits, and words). In the letter-updating task, stimuli were consonants with a minimum alphabetic distance of 2 between to-be-encoded letters (to avoid sequences like X-Y-Z). In the digit-updating task, stimuli were single-digit numbers (i.e., 1 to 9); the minimum numerical distance between to-be-encoded items was 2. In the word-updating task, stimuli were neutral, monosyllabic words taken from the online version of the MRC Psycholinguistic Database (Coltheart, 1981; Wilson, 1988; http://websites. psychology.uwa.edu.au/school/MRCDatabase/mrc2.html). Selected words were 2 to 5 letters

in length, with a Kucera-Francis frequency $f \geq 50$ (least and most frequent words, respectively, were *chain*, $f = 50$, and *are*, $f = 4393$). All updating tasks were programmed using MatLab and the Psychophysics Toolbox [version 2.54, 51] and run on IBM compatible computers. For each task, participants were instructed to be as fast and as accurate as possible. Additional details regarding the number of trials and block structure are provided in Table 1.

An example trial sequence for the digit updating task is illustrated in Fig 1. Each trial started with the presentation of a fixation cross presented centrally for 1000 ms. Then, three items were concurrently presented for 2000 ms in a single row of three black, rectangular frames (one item per frame). This was followed by a series of updating steps. On each updating step, one of the frames was cued by turning bold and red, indicating the item that was about to be replaced by a new item. The cue was presented for either a short (200 ms) or a long (1,500 ms) CTI (to achieve a constant inter-item interval, the CTI was preceded by an interval of blank frames that lasted 1,800 ms vs. 500 ms in the short vs. long CTI condition, respectively). Then, the new item was presented in the cued frame until a response was made or the maximum response time of 5,000 ms had elapsed. Participants were asked to press the space bar to indicate that they had encoded the new item and successfully updated their WM set. The number of updating steps per trial varied between 1 and 21, with a constant 10% stopping probability after each updating step (i.e., there was an average of ~9 updating steps per trial). The number of updating steps per trial was therefore unpredictable, and there was an equal incentive to undertake each updating step independent of the elapsed duration of the trial. After the updating phase, following a 500 ms interval of blank frames, participants were asked to recall the most recent item for each frame, thereby recalling a WM set of three items. To this end, a recall prompt (a blue question mark for the letter and digit tasks, and three blue underscores for the word task) appeared in each frame in random order. Participants typed the recalled items; that is, they typed the last recalled letter or digit for the letter and digit tasks, respectively, and the first three letters of the last recalled word in the word task (e.g., response of h-o-u would

**Table 1. Block order and number of trials per block for each task.**

| Block order | Trial type/Task | Number of trials per block |
|---|---|---|
| Updating (letter, digit, and word) | | |
| 1 practice block | - | 2 |
| 1 experimental block | - | 12[a] |
| Number Stroop and arrow flanker | | |
| 1 practice block | incongruent, congruent and neutral trials[b] | 24 |
| 3 experimental blocks | incongruent, congruent and neutral trials[b] | 72 (plus 2 warm-up trials) |
| Global-local and negative compatibility | | |
| 1 practice block | incongruent, congruent and neutral trials[b] | 24 |
| 2 experimental blocks | incongruent, congruent and neutral trials[b] | 72 (plus 2 warm-up trials) |
| Simon | | |
| 1 practice block | incongruent and congruent trials[b] | 24 |
| 3 experimental blocks | incongruent and congruent trials[b] | 48 (plus 2 warm-up trials) |
| Antisaccade | | |
| 1 experimental block | prosaccade trials | 40 (plus 10 warm-up trials) |
| 1 practice block | antisaccade trials | 24 |
| 1 experimental block | antisaccade trials | 48 (plus 2 warm-up trials) |

*Note*. In the analyses, only the (non-warm-up) trials from the experimental blocks were analyzed.

[a]For each updating task, there were approximately 108 updating steps in total (54 per cue-target interval condition).

[b]All trial types occurred with equal frequency in each block.

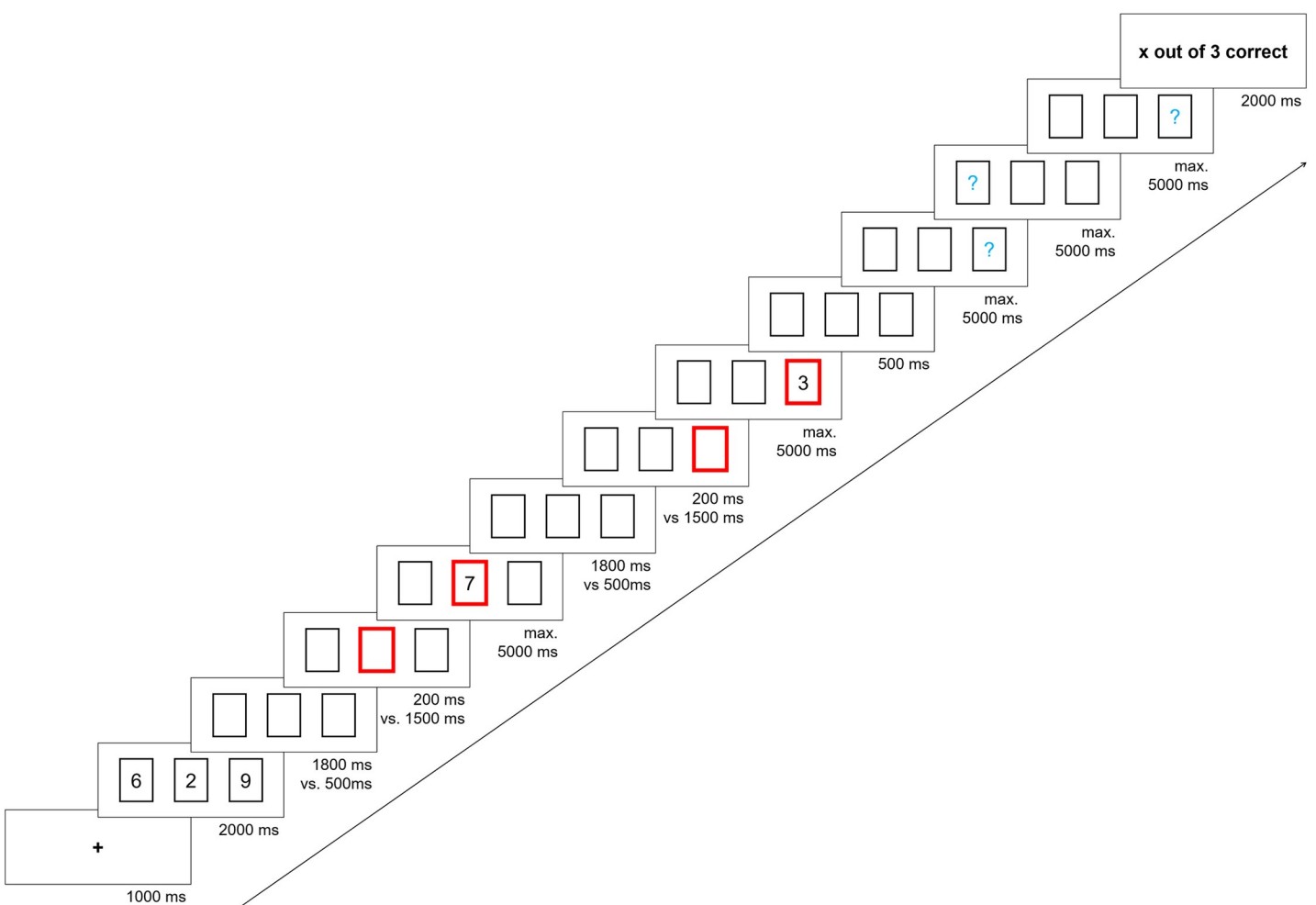

**Fig 1. Example of one trial sequence for the digit-updating task.** The duration of the blank at the beginning of each updating step (1800 ms or 500 ms) was determined by the length of the subsequent CTI: If the CTI was short (200 ms), the blank was long (1800 ms), if the CTI was long (1500 ms), the blank was short (500 ms). Thus, the retention interval between updating steps was constant at 2000 ms.

represent the recalled word 'house'). Each recall prompt was presented until a response was made or until 5,000 ms had elapsed.

**Inhibition tasks.**   All inhibition tasks were programmed using Tscope5 [52]. For each task, participants were instructed to be as fast and as accurate as possible. Feedback on accuracy was provided after each response; this was a smiley face for a correct response, and a frowny face for an incorrect response. All participants received the same pseudorandom trial sequence.

As episodic memory and associative learning impacts performance on inhibition tasks [e.g., 53, 54], three constraints were applied to reduce the influence of memory contributions for each task. First, trial-to-trial episodic memory was reduced by implementing trial sequences with no identical trial repetitions (see description of the Simon task for an exception). Second, associative learning was minimized by counterbalancing trial types (e.g., incongruent, congruent, and neutral; see below), response keys, presentation location, and individual stimulus exemplars as much as possible. Third, carry-over effects of any learning between tasks were reduced by avoiding overlap in stimulus materials across tasks. In addition, demands on

procedural working memory were also minimized by presenting the stimulus-response mappings on the lower part of the screen during the complete trial sequence.

For the congruency tasks (i.e., the number-Stroop, arrow-flanker, global-local, Simon, and negative-compatibility tasks), there were two to three trial types: *incongruent* trials (i.e., trials including conflict between relevant and irrelevant stimuli or responses features), *congruent* trials (i.e., trials without conflicting stimuli or responses features), and *neutral* trials (i.e., trials with only one response-relevant feature). Inhibition is assumed to occur in incongruent trials in order to reduce the interference triggered by irrelevant features. For each task, all trial types were presented randomly and occurred with equal frequency. First-order transitions between these trials also occurred with equal frequency. The main dependent measure was the congruency effect. This was calculated by subtracting RTs on congruent trials from RTs on incongruent trials (see the number-Stroop task below for an example, and the negative-compatibility task for an exception).

For most tasks (i.e., the number-Stroop, arrow-flanker, global-local, and Simon tasks), the trial structure was similar (see the negative-compatibility and antisaccade tasks for specific trial structure). That is, a central fixation cross was presented for 500 ms. Then, the stimulus was presented until a response was made or 2,000 ms had elapsed. Finally, accuracy feedback was presented centrally for 500 ms, followed by a 500 ms blank screen. An example trial sequence for each inhibition task is illustrated in Fig 2. Additional details regarding trial sequences and block structure are provided in Table 1. We next describe each task in more detail.

In the *number-Stroop* task, participants were presented with a centrally-displayed row of 1 to 4 identical digits (e.g., 222), and were instructed to indicate the number of digits presented while ignoring their numerical value [e.g., 11, 55]. There were three types of trials. In incongruent trials, the number of digits did not match the digits' numerical value (e.g., 33). In congruent trials, the number of digits matched the digits' numerical value (e.g., 333). In neutral trials, symbols (i.e., stimuli without a numerical value) were presented instead of digits (e.g., $ $ $). Participants responded by pressing the corresponding number keys (*1*, *2*, *3*, or *4*) along the upper row of a QWERTY keyboard with the index and middle fingers of the left and right hands, respectively.

In the *arrow-flanker* task, participants were presented with a centrally placed arrow among four flanking arrows (two on each side), arranged in a single row. They were instructed to indicate the direction of the central arrow (i.e., whether it pointed to the left or right) while ignoring the flanker arrows [e.g., 11, 25]. In incongruent trials, the central and flanking arrows pointed in opposite directions (e.g., ←←→←←); in congruent trials, central and flanking arrows pointed in the same direction (e.g., →→→→→); in neutral trials, hyphens were displayed as flankers instead of arrows (e.g.,–– → ––). Participants pressed the *A* and *L* keys with their left and right index fingers, respectively, to provide their responses.

In the *global-local* task, participants were presented on each trial with a large letter that was composed of smaller letters was presented centrally. Only the letters Y and V were used. Participants were instructed to perform the "local" task variant, that is, to identify the small constituent letters while ignoring the large letter [e.g., 11, 56]. In incongruent trials, the large letter did not match the small letters (e.g., a large Y made up of small v's); in congruent trials, the large letter matched the small letters (e.g., a large Y made up of small y's); in neutral trials, an unrelated letter was presented as the large letter (i.e., a large Z made up of either small v's or y's). Participants responded by pressing the *A* and *L* keys, respectively, for y and v with their left and right index fingers.

In the *Simon* task, participants were presented on each trial with a target shape (i.e., a black square or circle) that was presented on either the left or right side of the screen. Participants

**Fig 2. Example of one trial sequence for each inhibition task used in the present study.** Indivi. = individually.

were asked to indicate whether the shape was a circle or a square, while ignoring the position of the stimulus on the screen [e.g., 11, 57]. In incongruent trials, the target was presented on the opposite side to its response key (e.g., a square presented on the right requiring a response with the *A* key positioned on the left); in congruent trials, the target was presented on the same side as the target's response key (e.g., a square presented on the left side requiring a response with the *A* key positioned on the left). Following standard procedure, no neutral trials were employed. As this reduced the number of stimulus exemplars to four (i.e., square or circle presented on left or right), stimulus repetitions were allowed to make the correct response on

each trial unpredictable. Participants were asked to press the *A* or *L* keys with the index fingers of their left or right hand, respectively.

In the *negative-compatibility* task, participants were presented with a centrally positioned visual prime (i.e., a left- or right-pointing double arrow head, << or >>), which was backward-masked to prevent conscious identification. They were then presented with the target stimulus, also a double arrow head (<< or >>), which appeared either in the upper or lower part of the screen. Participants were instructed to indicate the direction of the target arrow head ("<<" or ">>") [e.g., 11, 58]. In incongruent trials, the target stimulus faced the opposite direction as the prime (e.g., <<, >>); in congruent trials, the target stimulus faced the same direction as the prime (e.g., >>, >>); in neutral trials, the prime was an unrelated symbol (i.e., an equal sign, =). In this task, it is assumed that the response induced by the prime is automatically suppressed and requires reactivation in order to respond to a congruent target [58, 59]. This results in *slower* responses on congruent trials compared to incongruent trials. Accordingly, the congruency effect was calculated by subtracting RTs on incongruent trials from RTs on congruent trials.

During each trial (see Fig 2), a fixation cross was presented centrally for either 250, 350, or 500 ms. This was followed by a blank screen lasting either 200 or 650 ms. Each time interval was selected randomly. The prime was presented for 30 ms and was followed by a 100 ms mask (the mask was created by superimposing the three prime symbols). Then, a blank screen appeared for 50 ms, followed by the target for 100 ms. Thus, the prime-target interval was 150 ms, which is a delay that typically produces negative-compatibility effects. This was followed by another blank screen presented until a response was made or 1,900 ms had elapsed. Participants were instructed to press the *A* or *L* key, using their left and right index fingers, respectively. Following the response, accuracy feedback was presented for 500 ms before another blank screen was presented for 500 ms prior to the commencement of the next trial.

In the *antisaccade* task, a visual cue (i.e., a small square) was presented on the left or right side of the screen. This was followed by the presentation of a target stimulus (i.e., an arrow pointing left, right, or upwards), which was backward-masked. Participants were instructed to indicate the direction of the target arrow [e.g., 11, 38]. This task had two types of trials. In antisaccade trials, the cue and the target appeared on opposite sides of the screen. In prosaccade trials, the cue and the target appeared on the same side of the screen. Antisaccade trials, thus, require inhibition of a reflexive saccade toward the cue to instead make a voluntary saccade away from the cue in order to identify the target. Participants responded by pressing the arrow key on the computer keypad that corresponded to the target, using the index, middle, and ring fingers of the right hand for left-, upwards, and right-pointing arrows, respectively.

During each trial (see Fig 2), a fixation cross appeared centrally for a variable amount of time (i.e., ranging from 1,500 to 3,250 ms in increments of 250 ms intervals, selected pseudo-randomly with no repetition). The cue was then presented for 166 ms, which was followed by the target arrow (presented inside a rectangle for an individually adapted time, as explained below) and the mask (created by superimposing the three target arrows), presented for 300 ms. Next, a blank screen appeared until a response was made or the maximum response time had elapsed (i.e., 1,700 ms minus the target presentation time of the current trial). Finally, accuracy feedback appeared for 500 ms, followed by a blank screen for 500 ms prior to the next trial.

Following precedent [11, 32], target presentation times (i.e., the time between target and mask onset) were calibrated based on individual responses. First, participants completed a block with prosaccade trials. In this block, target presentation times were adjusted so participants achieved 80% accuracy. To this end, the initial target presentation time of 150 ms was decreased by 17 ms after each correct response (down to a minimum of 34 ms), and increased by 85 ms after each incorrect response (up to a maximum of 740 ms). Participants then

completed a practice block and an experimental block with antisaccade trials. In these blocks, target presentation time was fixed individually to the median target presentation time of the calibration block (see Table 1). The dependent measure in this task was the difference in error rates between prosaccade and antisaccade trials.

## Procedure

The study was approved by the ethics committee of the University of Western Australia (approval number: RA/4/1/7133), and all participants provided written informed consent after receiving an approved information sheet. Participants first completed a demographic questionnaire, before completing the three updating tasks (in the order of letters, digits, and words) and finally the inhibition tasks (in the order number-Stroop, antisaccade, Simon, negative-compatibility, local, and arrow-flanker task). Participants were tested individually in a single testing session lasting approximately 90 minutes. Breaks were included at the end of every block of trials.

## Data preparation

The antisaccade task was the only task that had error rate as the primary dependent measure. All other tasks had RT as the primary dependent measure. RTs from anticipatory responses (i.e., RTs lower than 300 ms) were removed prior to analysis. For the inhibition tasks, RTs associated with incorrect responses were also excluded. Further, all tasks with RTs as the dependent measure excluded RTs falling 3 $SD$ above or below an individual's mean per condition.

For the updating tasks, we followed previous research [3, 5, 9] by computing a removal score as individual residuals from a simple linear regression model predicting the $RT_{short\ CTI}$ from the $RT_{long\ CTI}$. For comparative purposes and consistency with previous studies [e.g., 11, 20, 31, 32], we also computed the score for each inhibition task as a regression residual. That is, for the number-Stroop, arrow-flanker, global-local, and Simon tasks, the residual score was calculated from a simple linear regression model predicting $RT_{incongruent}$ from $RT_{congruent}$. For the negative-compatibility task, the residual score was computed from a simple linear regression model predicting $RT_{congruent}$ from $RT_{incongruent}$. For the antisaccade task, the residual score was computed from a simple linear regression model predicting $RT_{antisaccade}$ from $RT_{prosaccade}$.

## Model estimation

Latent variable models were estimated in R [60] using the lavaan package [61]. Model fit was evaluated via multiple fit indices [62, 63]: the $\chi^2$ goodness-of-fit statistic, Bentler's comparative fit index (CFI), root mean square error of approximation (RMSEA), standardized root-mean-square residual (SRMR), Akaike information criterion (AIC), and the Bayesian information criterion (BIC). For the $\chi^2$ statistic, a small, non-significant value indicates a good fitting model. For the CFI, values between .90 and .95 indicate acceptable fit, and values larger than .95 indicate good fit. RMSEA values smaller than .06 and SRMR values smaller than .08 also indicate good fit. It is noted that the RMSEA statistic tends to over-reject true population models at small sample sizes (i.e., smaller than 250 [see 62]); however, RMSEA was reported for the sake of completeness. Smaller AIC and BIC indices indicate better fitting models.

Two analyses were performed in order to test if one model fit the data better than another. First, we conducted $\chi^2$ difference ($\Delta\chi^2$) tests on nested models. If the more complex model (i.e., the model with more free parameters) yields a reduction in $\chi^2$ that is significant given the loss of degrees of freedom, it is considered a better fitting model. Second, in order to assess the

strength of evidence for both the null and alternative hypotheses, Bayesian hypothesis tests using BIC approximation [64] were performed. To calculate a Bayes factor (BF) showing evidence in favor of the null hypothesis ($BF_{01}$) and evidence in favor of the alternative hypothesis ($BF_{10}$), we used the difference between the BIC for the null hypothesis (e.g., the single-factor model) and the BIC for the alternative hypothesis (e.g., the 2-factor model). In the present study, all BF were interpreted following Raftery's [65] classification scheme. That is, BF values between 1 and 3 were considered weak evidence, values between 3 and 20 were considered positive evidence, values between 20 and 150 were considered strong evidence, and values larger than 150 were considered very strong evidence.

In addition, the following criteria had to be met for a model to be considered an adequate representation of a latent variable: (1) the Kaiser-Meyer-Olkin (KMO) index–a measure of whether the correlation matrix is factorable–should be larger than .60 [66]; (2) most of the error variances needed to be lower than .90; (3) most of the factor loadings had to be significant and larger than .30; (4) no factor should be dominated by a large loading from one task; (5) the amount of shared variance across tasks—that is, "factor reliability" as assessed by coefficient ω [67]—had to be high (i.e., about .70). The data, scripts analyses and results presented in this work can be found at https://osf.io/c8qb2.

## Results

Results are reported in two sections. First, reliabilities and the correlational patterns of the scores derived from all tasks are presented. Second, CFA results are presented. Each construct was measured at the latent-variable level before examining the potential relationship between the interference-control processes assessed in removal and inhibition tasks.

### Reliabilities and correlations

As shown in Table 2, the reliability estimates (split-half) varied widely. In particular, the reliability estimate of the number-Stroop task measure was considered too low (ρ = .10) and,

**Table 2. Descriptive statistics.**

| Construct/Task | *M* | *SD* | Min. | Max. | Skew | Kurtosis | Reliability |
|---|---|---|---|---|---|---|---|
| Inhibition | | | | | | | |
| Number Stroop | -2.17 | 24.39 | -73.02 | 74.65 | 0.34 | 0.79 | .10 |
| Arrow flanker | -0.38 | 19.90 | -62.67 | 51.92 | 0.11 | 0.96 | .66 |
| Local | 0.90 | 33.09 | -72.48 | 115.70 | 0.56 | 1.14 | .48 |
| Simon | -0.80 | 29.35 | -92.04 | 78.38 | -0.31 | 0.77 | .73 |
| Negative compatibility | -1.07 | 23.21 | -95.43 | 54.65 | -0.45 | 1.85 | .56 |
| Antisaccade | -0.87 | 20.75 | -38.18 | 45.14 | 0.20 | -1.02 | .92 |
| Removal | | | | | | | |
| letter updating | 4.27 | 136.18 | -284.18 | 476.44 | 1.29 | 2.36 | .68 |
| digit updating | 3.11 | 133.78 | -334.23 | 420.40 | 0.64 | 1.33 | .69 |
| word updating | 7.04 | 131.53 | -320.48 | 457.31 | 0.64 | 0.68 | .58 |

*Note*. For the number-Stroop, arrow-flanker, global-local, and Simon task, the scores were computed as residuals from a simple linear regression model predicting the $RT_{incongruent}$ from the $RT_{congruent}$. For the negative-compatibility task, the residual scores were computed from a simple linear regression model predicting the $RT_{congruent}$ from the $RT_{incongruent}$. For the antisaccade task, the residual scores were computed from a simple linear regression model predicting the $RT_{antisaccade}$ from the $RT_{prosaccade}$. For the three updating tasks, the scores were computed as residuals from a simple linear regression model predicting the $RT_{short\ CTI}$ from the $RT_{long\ CTI}$. Reliabilities were calculated by adjusting split-half correlations with the Spearman–Brown prophecy formula. Split-half correlations were computed between odd and even items. Min. = minimum; Max. = maximum.

therefore, not included in the CFA analyses. The remaining estimates evidenced reliability of essentially half their variance or more.

Correlations are presented in Table 3. To calculate the weight of evidence for each correlation, we computed Bayes factors ($BF_{01}$, in favor of the absence of a correlation; $BF_{10}$, in favor of the correlation); these are presented in S1 Table and interpreted following Raftery's [65] classification scheme. Correlations between removal measures were strong and significant, with all $BF_{10}$ suggesting very strong evidence for the correlations. This indicates that the removal tasks measured a common removal ability; the fact that they did not correlate perfectly indicates task-specific variance. In contrast, most of the correlations between the inhibition measures were low and non-significant. This indicates that the inhibition tasks were not measuring a common inhibition ability. In addition, the correlations between the inhibition and the removal measures were mostly weak and non-significant. This was further supported by the $BF_{01}$ analysis, which suggested positive to strong evidence for the absence of correlations between the removal measures and the inhibition measures.

## Confirmatory factor analysis

Using CFA, we first corroborated the construct of removal as a latent variable. Next, we aimed to observe a coherent factor of inhibition, in order to then attempt to examine the relationship between the interference-control processes assessed in removal and inhibition tasks. Goodness-of-fit statistics of the models are presented in Table 4. Models are depicted in Fig 3.

### Removal latent variable

To establish removal as a latent variable, we fit a saturated model (Model 1), in which all three removal measures loaded onto a single factor. As depicted in Fig 3A, all tasks had significant,

**Table 3. Pearson correlation coefficients.**

|  | Number Stroop | Arrow flanker | Local | Simon | Neg. comp. | Antisaccade | Letter upd. | Digit upd. |
|---|---|---|---|---|---|---|---|---|
| Arrow flanker | .08 | | | | | | | |
|  | [-.10, .26] | | | | | | | |
| Local | -.10 | -.05 | | | | | | |
|  | [-.27, .07] | [-.27, .16] | | | | | | |
| Simon | .01 | .04 | -.17 | | | | | |
|  | [-.23, .25] | [-.15, .23] | [-.38, .06] | | | | | |
| Neg. comp | .03 | -.02 | .08 | .03 | | | | |
|  | [-.13, .21] | [-.23, .18] | [-.10, .27] | [-.14, .22] | | | | |
| Antisaccade | .08 | .07 | -.01 | .05 | .15 | | | |
|  | [-.10, .26] | [-.12, .25] | [-.18, .17] | [-.13, .23] | [-.02, .32] | | | |
| Letter upd. | .15 | -.06 | -.22* | .13 | -.01 | .10 | | |
|  | [-.01, .30] | [-.23, .11] | [-.42, -.03] | [-.08, .35] | [-.24, .22] | [-.10, .31] | | |
| Digit upd. | .16 | -.03 | -.11 | .03 | .12 | .04 | **.40*** | |
|  | [-.03, .35] | [-.20, .13] | [-.27, .05] | [-.21, .25] | [-.04, .28] | [-.15, .23] | [.25, .56] | |
| Word upd. | .12 | .10 | .18 | .03 | .03 | .05 | **.39*** | **.43*** |
|  | [-.08, .32] | [-.07, .28] | [-.002, .36] | [-.17, .22] | [-.19, .24] | [-.13, .23] | [.18, .61] | [.28, .58] |

*Note.* Ninety-five % bootstrapped confidence intervals (10000 random samples) are presented in brackets. Correlations for which the Bayes factor suggested positive to strong evidence for the null hypothesis ($BF_{01}$) are presented in italics; correlations for which the Bayes factor suggested positive to strong evidence for the alternative hypothesis ($BF_{10}$) are presented in bold. Bayes factors were estimated in R [60] using the BayesMed package [68] with default prior scales. Bayes factors for each correlation are presented in S1 Table. Neg. comp. = Negative compatibility; upd. = updating.

* $p < .05$.

**Table 4. Goodness-of-fit statistics.**

| Construct/Model | KMO | $\chi^2$ | df | p | CFI | RMSEA [90% CI] | SRMR | AIC | BIC |
|---|---|---|---|---|---|---|---|---|---|
| Removal | | | | | | | | | |
| Model 1 (saturated) | .66 | 0 | 0 | - | 1 | 0 [0, 0] | 0 | 4128.15 | 4144.36 |
| Inhibition | | | | | | | | | |
| Model 2 | .50 | 4.45 | 5 | .486 | 1 | 0 [0, .13] | .05 | 5102.73 | 5129.74 |
| Model 4 (saturated) | .51 | 0 | 0 | - | 0 | 0 [0, 0] | 0 | 3045.10 | 3061.30 |
| Model 5 | .87 | 4.45 | 5 | .486 | 1 | 0 [0, .13] | .05 | 5574.38 | 5601.39 |
| Model 7 (saturated) | .80 | 0 | 0 | - | 0 | 0 [0, 0] | 0 | 3263.01 | 3279.22 |
| Each inhibition measure was the predictor of the removal factor | | | | | | | | | |
| Model 8: Antisaccade | .66 | 2 | 0.47 | .790 | 1 | 0 [0, .12] | .02 | 5107.74 | 5126.65 |
| Model 8: Simon | .66 | 2 | 1.41 | .494 | 1 | 0 [0, .17] | .03 | 5184.10 | 5203.01 |
| Model 8: Negative compatibility | .65 | 2 | 1.57 | .456 | 1 | 0 [0, .18] | .03 | 5132.58 | 5151.48 |
| Model 8: Local | .55 | 2 | 55.80 | < .001 | .10 | .49 [.39, .61] | .23 | 5249.81 | 5268.72 |
| Model 8: Arrow flanker | .64 | 2 | 50.15 | < .001 | 0 | .47 [.36, .58] | .22 | 5146.51 | 5165.41 |

*Note.* Model 1 = Single-factor model in which all removal measures loaded on a factor; Model 2 = Single-factor model in which all inhibition tasks loaded on a factor; Model 4 = Single-factor model in which only the antisaccade, Simon, and negative-compatibility tasks were included; Model 5 = same model as Model 2, except that the correlation was disattenuated for imperfect reliability; Model 7 = same model as Model 4, except that the correlation was disattenuated for imperfect reliability; Model 8 = Single-factor model in which the regression between each inhibition measure and the removal factor was freely estimated. KMO = Kaiser-Meyer-Olkin index for the correlation matrix; CFI = comparative fit index; RMSEA = root mean square error of approximation; CI = confidence interval; SRMR = standardized root-mean-square residual; AIC = Akaike information criterion; BIC = Bayesian information criterion.

moderate, positive standardized loadings, and error variances were relatively low. Factor reliability was high, ω = .67. Taken together, these findings show a coherent factor of removal, consistent with previous findings [3, 5, 9].

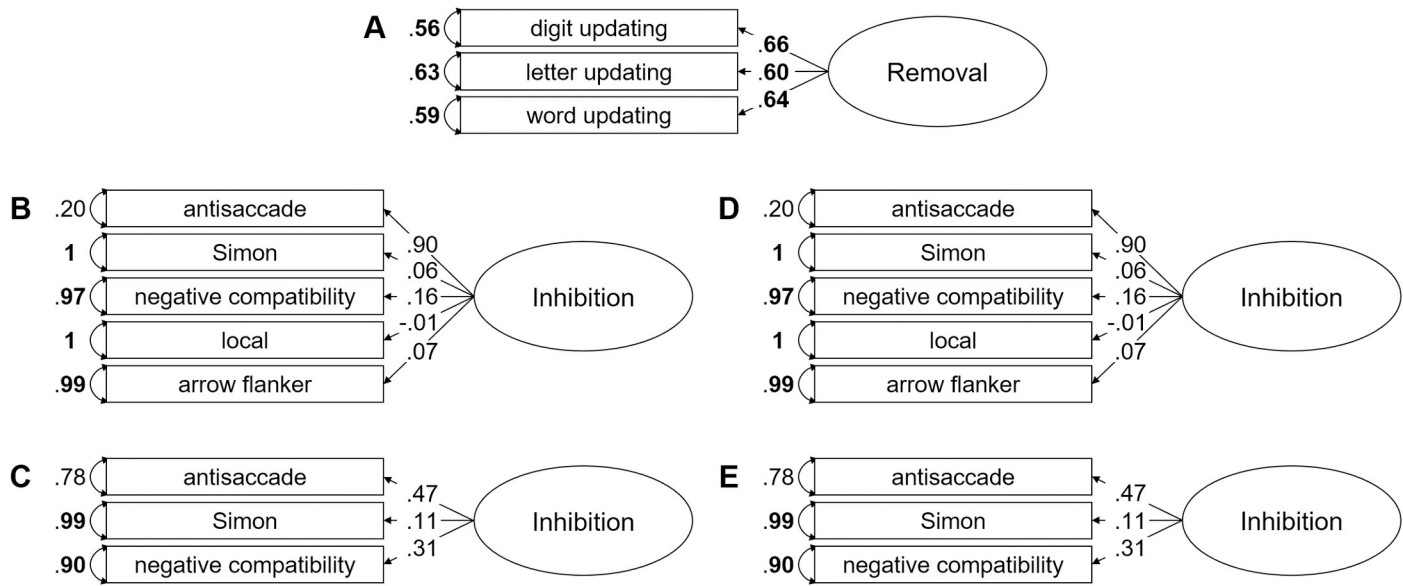

**Fig 3. Illustration of the different models computed in the present study.** (A) Single-factor model with removal as a latent variable (Model 1). (B) Single-factor model in which all tasks assumed to assess inhibition loaded on a latent variable (Model 2). (C) Single-factor model in which the antisaccade, Simon and negative-compatibility tasks loaded on a latent variable (Model 4). (D) Same model as Model 2, except that the correlation was disattenuated for imperfect reliability (Model 5). (E) Same model as Model 4, except that the correlation was disattenuated for imperfect reliability (Model 7). The numbers next to the straight, single-headed arrows are the standardized factor loadings (interpretable as standardized regression coefficients). The numbers adjacent to the curved, double-headed arrows next to each task are the error variances, attributable to idiosyncratic task requirements and measurement error. For all parameters, boldface type indicates *p* < .05.

**Inhibition latent variable.**   The next step was to establish a coherent factor of inhibition. To this end, we first fit Model 2, in which the tasks assumed to measure inhibition loaded onto a single factor. As shown in Table 4, this model had acceptable fit statistics. However, as illustrated in Fig 3B, one loading was substantially higher than the others (i.e., the loading for the antisaccade measure). Thus, the latent variable from Model 2 represented mainly the variance from one measure. Second, for most measures, the factor loadings were non-significant and the error variances very high. This indicated that a large proportion of variance from the inhibition measures could not be accounted for by a latent inhibition variable. Accordingly, the factor reliability for this latent variable was low, ω = .18. Together, this suggests that the model had low explanatory power.

To identify a coherent factor of inhibition, we also fit the following models: (1) a unitary model (Model 3) which excluded the arrow-flanker task, because this task has been conceptualized in the literature as a task that measures the ability to ignore irrelevant information from the environment, rather than the ability to suppress distracting information [e.g., 38]; (2) a unitary model (Model 4) which excluded the local task, because this task had the lowest reliability among the remaining inhibition measures (see Table 2); (3) a Model 5 which was similar to Model 2, except that the correlation matrix was disattenuated for imperfect reliability by applying the following formula for each correlation [see 42]: correlation / SQRT (reliability of the first measure × reliability of the second measure); (4 and 5) two models that are similar to Models 3 and 4, respectively, except that each correlation matrix was disattenuated for imperfect reliability by applying the formula presented above (Models 6 and 7).

Models 3 and 6 did not converge. Model 5 had the same issues as Model 2; that is, although Model 5 had acceptable fit statistics, the loading for the antisaccade measure was substantially higher than the others (see Fig 3D), the factor loadings were non-significant and the error variances very high. Accordingly, the factor reliability for this factor was low, ω = .10. This suggests that the latent variable from Model 5 represented mainly the variance from one measure and a large proportion of variance from the measures could not be accounted for by the inhibition factor, thus emphasizing the low explanatory power of this model. Models 4 and 7 are saturated models (see Table 4). However, for these models, the factor loadings were non-significant. In addition, error variances were high (see Fig 3C and 3E). Factor reliability was also low, ω = .20 and ω = .18, respectively.

Together, these results are consistent with the assumption of a common factor, which however explained only a small proportion of the variance from each task. Moreover, taking into account the reliability when computing the correlations improved the KMO index for the correlation matrix (see Table 4). However, this optimization in the KMO index did not result in a coherent factor of inhibition. Consequently, an inhibition latent-variable model was not specified, and it was determined that conducting a latent-variable analysis was not appropriate for the inhibition data.

**The relation between inhibition measures and removal.**   As inhibition could not be identified at the latent-variable level, the relationship between the interference-control processes assessed in removal and inhibition tasks was investigated adopting a different strategy. That is, the relationship of each individual measure of inhibition with the removal factor was examined separately. To this end, we fit a series of models in which each inhibition measure independently predicted the removal factor (Model 8). Goodness-of-fit statistics are summarized in the lower part of Table 4. The model fit to the data was acceptable for only three inhibition measures (i.e., the antisaccade, Simon, and negative-compatibility measures). However, the regression coefficients between the individual measures of inhibition and the removal latent variable were small and non-significant (antisaccade: .10, p = .389; Simon: .09, p = .416; and negative compatibility: .09, p = .425). Moreover, the strength of each regression coefficient

was assessed by comparing each of these models to a model that set regression to the removal factor to 0 (Model 9). Using Bayesian hypothesis testing with BIC approximation, it was found that the $BF_{01}$ in favor of the null hypothesis (i.e., Model 8 = Model 9) provided positive evidence against a correlation between each inhibition measure and the removal factor ($BF_{01}$ = 7.25, $BF_{01}$ = 7.57, and $BF_{01}$ = 7.96 for the antisaccade, Simon, and negative-compatibility tasks, respectively). Together, these findings suggest that the individual measures of inhibition were not related to removal ability.

## Multiverse-analysis approach

To test the robustness of our results, we re-ran the analyses by separately applying the following modifications. (1) For the inhibition measures, we computed two alternative dependent measures. That is, we calculated the interference effect (i.e., the difference between incongruent and neutral trials for the number-Stroop, arrow-flanker, and local tasks, and the difference between congruent and neutral trials for the negative-compatibility task) as well as the facilitation effect (i.e., the difference between congruent and neutral trials for the number-Stroop, arrow-flanker, and local tasks, and the difference between incongruent and neutral trials for the negative-compatibility task). (2) Instead of using residuals as dependent measures, we computed proportional scores for all measures (i.e., [$RT_{short\ CTI}$—$RT_{long\ CTI}$] / $RT_{short\ CTI}$ for the three updating tasks, [$RT_{incongruent}$—$RT_{congruent}$] / $RT_{incongruent}$ for number-Stroop, arrow-flanker, local, and Simon tasks, [$RT_{congruent}$—$RT_{incongruent}$] / $RT_{congruent}$ for the negative-compatibility task, and [$RT_{antisaccade}$—$RT_{prosaccade}$] / $RT_{antisaccade}$ for the antisaccade task). (3) For each construct (inhibition and removal), we estimated bi-factor models. That is, for removal, we fit a bi-factor model, which forced performance from both CTI conditions to load on a baseline factor, and performance from the short CTI condition to load on a removal factor. For inhibition, the bi-factor model forced performance on the antisaccade and prosaccade trials of the antisaccade task, and performance on incongruent and congruent trials of the number-Stroop, arrow-flanker, Simon, local, and negative-compatibility tasks to load on a baseline factor, and performance on antisaccade trials of the antisaccade task, congruent trials of the negative-compatibility task, and incongruent trials of all other tasks to load on an inhibition factor. (4) We included participants who had missing data in less than half of the inhibition measures, and we ran the CFA with case-wise maximum-likelihood estimation. The goodness-of-fit statistics of all these model assessments and the parameter estimates can be found at https://osf.io/c8qb2. The results of these supplementary tests were either equivalent to those presented in this article or worse (e.g., the model(s) had worse fit statistics or the model(s) did not converge).

## Discussion

The purpose of the present study was to investigate the extent to which the tasks used to assess removal and inhibition measure the same cognitive process. Participants completed three tasks assessing removal efficiency (i.e., a modified updating task using either letters, digits, or words as stimuli) and six tasks designed to assess inhibition (i.e., the number-Stroop, arrow-flanker, global-local, Simon, negative-compatibility, and antisaccade tasks). The removal measures displayed good reliability estimates and high correlations. Bayesian inference provided very strong evidence in favor of the interrelations. Using CFA, a coherent latent factor of removal was established, replicating previous findings [3, 5, 9]. In contrast, although most inhibition measures had acceptable reliabilities, the inhibition measures did not correlate with each other or the removal measures. Bayesian inference provided positive to strong evidence for the absence of correlations. CFA was not able to identify a model with inhibition as a

coherent latent variable. Given these difficulties, we investigated the relation between the inter-ference-control processes assessed in inhibition and removal tasks by examining the relations between the individual measures of inhibition and the latent variable of removal. The results showed that none of the inhibition measures was related to the removal factor, suggesting these measures assess functions that are unrelated to removal ability.

## The difficulty of establishing inhibition as a latent variable

In the present study, the internal consistency reliabilities of most tasks were similar to Rey-Mer-met and colleagues [11], where the same inhibition measures were used, despite participants performing fewer trials in the present study (e.g., in the arrow-flanker and antisaccade tasks). There were, however, two exceptions. The number-Stroop and the negative-compatibility tasks had lower reliabilities in the present study, whereas these measures provided acceptable to good reliability estimates (.71 and .85, respectively) in Rey-Mermet and colleagues [11]. The decrease in reliability for the number-Stroop task may be the result of the present study having fewer tri-als, suggesting good reliability estimates require a different number of trials across inhibition measures. For the negative-compatibility task, participants performed the same number of trials in both studies. The only difference is that in Rey-Mermet and colleagues [11], participants per-formed a positive-compatibility task before and after the negative-compatibility task. The posi-tive-compatibility task was implemented similarly to the negative-compatibility task, except the prime-target interval being 0 ms (i.e., no blank screen between prime and target); this task pro-vided typical congruency effects (i.e., an RT cost in incongruent trials). Therefore, the present results in conjunction with the results of Rey-Mermet and colleagues [11] suggest that complet-ing the positive-compatibility task before the negative-compatibility task may potentially affect reliability estimates for the negative-compatibility task. This may occur via reinforcement of the mapping between each stimulus exemplar and its response.

Overall, inhibition measures did not have any significant correlations between them. In addition, a coherent latent construct of inhibition could not be established. Even when imper-fect reliability estimates were taken into account by disattenuating the correlation matrix, we were unable to establish a coherent inhibition construct. These results are in line with recent studies highlighting the difficulty of establishing inhibition as a coherent latent construct [11, 32, 39, 69]. In particular, they are in line with the studies in which the inhibition factor does not represent much common variances across the different measures because it had very large residual variances [e.g., 11, 20, 27, 29, 31] and one high factor loading dominating the remain-ing low factor loadings [see, e.g., 11, 20, 23, 24, 31, 36].

As pointed out recently by Draheim and colleagues [69, 70], one possible reason for the dif-ficulty in establishing a coherent factor of inhibition might be the use of RTs—and in particu-lar, RT difference scores—because these scores have been associated with low reliability and speed-accuracy trade-offs. Besides the substantial impact of measurement error [71], previous research has emphasized the low performance variability in the tasks used to assess inhibition, resulting in a low test/re-test reliability of these measures [see 40, 41]. However, both the pres-ent study and Rey-Mermet and colleagues [11] showed that when computed with a sufficient number of trials, RT difference scores can be measured with reasonable internal consistency reliabilities. Perhaps more importantly, in the present study we also disattenuated the correla-tions for imperfect reliability in the inhibition measures. Moreover, when using a calibration procedure in order to measure inhibition through accuracy, thus pushing speed-accuracy trade-offs into accuracy scores, Rey-Mermet and colleagues [32] were still unable to establish a factor of inhibition. Thus, the use of RTs and RT difference scores per se cannot be the sole reason for the difficulties to establish inhibition at the latent-variable level.

A further reason why the use of difference scores may impede extraction of a latent inhibition variable is that computing difference scores is based on the premise of additive factors [32, 69]. However, this premise is questionable because it is not known whether the durations of baseline processes and inhibitory processes combine additively to the RTs of incongruent trials. It is for this reason that both the present study and Rey-Mermet and colleagues [11, 32] applied a bi-factor modeling approach to overcome the premise of additive factors. In this approach, all incongruent and congruent trials were forced to load on a baseline factor and the incongruent trials were forced to load on an inhibition factor. However, even when using a bifactor model, no inhibition-specific variance could be established. Thus, the use of difference scores—either in RTs or accuracy rates—can also not totally explain the difficulty of establishing an inhibition factor at the latent-variable level.

Thus, it seems that there remains only one reason to explain the failure to establish a coherent inhibition latent factor. It is possible that the selected tasks used to measure inhibition do not assess a common underlying construct, but the highly task-specific ability to reduce the interference arising in that particular task. As interference from a flanking arrow differs, for example, from interference from an irrelevant stimulus location (as in the Simon task), the abilities to minimize interference in these tasks may be task-specific. Thus, whatever is measured by the tasks that are used to assess inhibition seems highly task-specific, which thus calls into question the view of inhibition as a psychometric construct [see 11].

## The relationship between removal and inhibition tasks

The present study's results showed the difficulty of establishing a coherent factor of inhibition. It was, therefore, not possible to determine the relationship between the interference-control processes assessed in inhibition and removal tasks at the latent-variable level. As the purpose of the present study was to determine the relation between the interference-control processes measured in inhibition and removal tasks, a different strategy was then adopted by investigating the relations between the individual measures of inhibition and the latent factor of removal. None of the inhibition measures related to the removal construct. This occurred even in tasks where the chance of finding a relation was optimized due to measures being on the same scale (i.e., RTs; all inhibition tasks except the antisaccade task). Thus, the results of the present study indicate that the inhibition measures assess interference-control processes that are different from removal ability. This first shows that although removal can be estimated at the latent-variable level in a valid and reliable way, removal tasks cannot be used to measure the interference-control processes assessed in inhibition tasks. Furthermore, this emphasizes the necessity of conceptualizing removal based on previous research and WM theory. One possibility is that removing an item from WM rather represents "unbinding" the item from the context in which it occurred (e.g., unlearning the association between the digit "6" and the frame in which it appeared) [see 2, 3, 5, 72]. In fact, this conceptualization maps onto the way removal has been implemented in computational WM models [8].

Together, the present results show that performance on the inhibition tasks we used relies on interference-control processes that are different from removal. On a general level, the results also provide additional evidence that we should stop using inhibition as an umbrella term for a variety of distinct processes. Thus, although the present study supports the notion that removal is not related to the interference-control processes assessed by the inhibition tasks, in order to progress, the field will need to specify how inhibition could act on irrelevant/outdated information in WM, and should attempt to build experimental paradigms that are able to disentangle inhibition from other mechanisms of conflict resolution (as has been done with inhibition in episodic memory [73, 74]).

## The impact of task selection

In the present study, we used tasks which were assumed to asses removal and inhibition, respectively. However, the reader may wonder whether our task selection can explain the present study's results. First, one may question our decision not to use the color Stroop task, the stop-signal task, or the sustained-attention-to response task (SART) task in accordance with previous research [e.g., 11, 13, 20, 27, 37, 38]. In the color Stroop task, participants are asked to respond to the color of color words while ignoring their meaning. In the stop-signal task, participants are asked to perform an ongoing decision task (e.g., deciding whether a picture represents a living or non-living object) unless a stop-signal occurs (e.g., the frame of the picture turns pink). In the SART task, participants are asked to perform a go/no-go task in which the goal is to respond to words from one category (e.g., animals) while withholding responses for words from another category (e.g., vegetables), which occur more rarely.

We did not include the color Stroop task because our testing would have required a key-press version of the task (i.e., participants would have been asked to press four response keys in order to respond to the colors red, blue, green or yellow). However, with such a task version, the set of stimulus-response mappings would have been large and each stimulus-response mapping would have been completely arbitrary. This would have increased WM requirements, thus possibly impacting the relation between task performance and WM. We also did not include the stop-signal task because participants have been found to wait for the stop signal to occur in order to stop their responses [e.g., 11, 38], meaning that strategical thinking can influence the inhibitory processes measured with that task. Although there exist versions of the stop-signal task that minimize the use of strategies (e.g., by implementing monetary incentives [see 75]), these versions have rarely been used in individual-differences research using a CFA and SEM approach [11, 13, 27, 30, 38 but see 32 for one exception]. As stated in the introduction, we opted for inhibition tasks that are broadly assumed to require inhibition and have been commonly used in individual-differences research using SEM. Similarly, we did not include the SART because the dependent measures typically used in this task (e.g., accuracy rates on no-go trials or the signal detection measure $d'$) do not control for variance in processing speed. If fast and slow individuals respond within about the same time, slower individuals will make more errors, thus transferring some speed variance onto accuracy and $d'$.

Second, one may argue that we found a coherent factor for removal but not for inhibition because the tasks used to assess removal were more similar than those used to asses inhibition. In other words, a coherent inhibition factor may have emerged had we used the same inhibition task with different stimulus materials. The Stroop task may be one candidate for such a design because there exists a color version (as described earlier), a number version (such as used in the present study) and a spatial version (in which participants indicate the left-right position of the words "left" or "right" while ignoring their meaning). However, previous research has shown only small to moderate correlations between the different versions of the Stroop task (i.e., -.22 –.38 for the correlation between color and number versions, .21 –.52 for the correlation between color and spatial versions, and .08 –.28 for the correlation between number and spatial versions) [see 11, 20, 27, 31, 32, 55, 76]. These studies included participants with a large range of ages (i.e., from 18 to 89 years old). When only young adults were considered, the range for the correlations decreased further (i.e., to the range of -.09 –.11) [see 11, 20, 27, 32]. This questions the extent to which the different versions of the Stroop task reflect the same construct, suggesting that even with a more balanced design, the results would be similar to those of the present study. Moreover, using a wide variety of tasks, such as we did in the present study, ensures that our investigation of the relation between removal and inhibition tasks tapped into the different ways inhibition is currently assessed.

Finally, as removal is a process that reduces WM interference, one may wonder whether it would have been easier to find a relation between removal and inhibition tasks if we had used tasks assessing memory interference (such as, e.g., the cued recall task, in which participants are presented with one or two lists of words and are asked to retrieve the word on the most recent list that belongs to a cued category, ignoring any previous list). However, as we aimed to determine whether the correlations between (1) WM and attentional control, and between (2) WM updating and inhibition were driven by a common removal/inhibition factor, it was imperative to use the same type of tasks as the research investigating these correlations. Moreover, it has been argued that memory-interference tasks involve delayed recall of long-term associations that are more robust than the temporary bindings created in WM for performing the inhibition and removal tasks [2]. Thus, memory-interference tasks may operate in a different memory system than the inhibition and removal tasks used in the present study (long-term memory vs. WM, respectively).

## Conclusion

To summarize, we investigated the extent to which the tasks used to assess removal and inhibition measure the same construct. Despite disattenuating the correlations for imperfect reliability, the measures of inhibition generally did not correlate amongst each other and also did not correlate with the removal measures. While a coherent factor of removal could be established, no factor of inhibition could be identified. Thus, to investigate the relation between the interference-control processes assessed in removal and inhibition tasks, we examined the relations between the individual measures of inhibition and the factor of removal. Irrespective of the tasks used to assess inhibition, the inhibition measures did not relate to the removal factor. Thus, these results question the assumption that removing no-longer information from WM is somehow related to the interference control assessed in inhibition tasks.

## Supporting information

**S1 Table. Bayes factors in favor of the alternative hypothesis ($BF_{10}$) and in favor of the null hypothesis ($BF_{01}$) for the Pearson correlation coefficients.**
(PDF)

## Acknowledgments

We thank Charles Hanich and Jiaxin Tay for their help in data collection.

## Author Contributions

**Conceptualization:** Alodie Rey-Mermet, Krishneil A. Singh, Gilles E. Gignac, Christopher R. Brydges, Ullrich K. H. Ecker.

**Data curation:** Alodie Rey-Mermet, Krishneil A. Singh, Ullrich K. H. Ecker.

**Formal analysis:** Alodie Rey-Mermet, Krishneil A. Singh.

**Funding acquisition:** Krishneil A. Singh.

**Investigation:** Krishneil A. Singh.

**Methodology:** Alodie Rey-Mermet, Krishneil A. Singh, Ullrich K. H. Ecker.

**Project administration:** Krishneil A. Singh, Ullrich K. H. Ecker.

**Resources:** Alodie Rey-Mermet, Krishneil A. Singh, Ullrich K. H. Ecker.

**Software:** Alodie Rey-Mermet, Krishneil A. Singh, Ullrich K. H. Ecker.

**Supervision:** Ullrich K. H. Ecker.

**Validation:** Alodie Rey-Mermet.

**Visualization:** Alodie Rey-Mermet.

**Writing – original draft:** Alodie Rey-Mermet, Krishneil A. Singh.

**Writing – review & editing:** Alodie Rey-Mermet, Krishneil A. Singh, Gilles E. Gignac, Christopher R. Brydges, Ullrich K. H. Ecker.

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
