## [Decision Letter · Decision Letter 0]

28 Jul 2020

PONE-D-20-13452

Removal of information from working memory is not related to inhibition

PLOS ONE

Dear Dr. Additional Editor Comments (if provided):

Dear Dr. Rey-Mermet,

Your paper has been reviewed by two expert referees and I have read through your paper carefully myself.

As you will see the reviews are somewhat mixed as they see some positive aspects but also highlight quite a number of critical points. Given the severity of some of the arguments presented regarding the conceptual rationale and the task selection, as well as the changes suggested in the discussion section, the current version of the manuscript cannot be accepted for publication in PLOSONE. However, as the reviewers also see some potential for a novel contribution I strongly suggest to resubmit your study.

Therefore, we invite you to submit a revised version of the manuscript that addresses the points raised during the review process. Please note that you need to convince reviewers and myself that you have managed to deal with the most important issues in a satisfactory manner. 

We look forward to receiving your revised manuscript.

Kind regards,

Erika Borella, Ph.D.

Academic Editor

PLOS ONE

Journal Requirements:

2. Your ethics statement must appear in the Methods section of your manuscript. If your ethics statement is written in any section besides the Methods, please move it to the Methods section and delete it from any other section. Please also ensure that your ethics statement is included in your manuscript, as the ethics section of your online submission will not be published alongside your manuscript.

Reviewers' comments:

Reviewer's Responses to Questions

**Comments to the Author**

1. Is the manuscript technically sound, and do the data support the conclusions?

Reviewer #1: Partly

Reviewer #2: Yes

2. Has the statistical analysis been performed appropriately and rigorously? 

Reviewer #1: Yes

Reviewer #2: Yes

3. Have the authors made all data underlying the findings in their manuscript fully available?

Reviewer #1: Yes

Reviewer #2: Yes

4. Is the manuscript presented in an intelligible fashion and written in standard English?

Reviewer #1: Yes

Reviewer #2: Yes

5. Review Comments to the Author

Reviewer #1: Brief Summary

The manuscript reports one study aimed to examine whether the ability to update information in working memory (as measured with three updating tasks) relates to the construct of inhibition (as measured with six very different tasks that some have claimed to tap into inhibition). While a latent variable of removal/updating was evident from the analyses, this not was the case for inhibition. Individual measures of inhibition did not even relate to removal as a construct. The authors conclude these results support the idea that removal is different from inhibition.

General evaluation and comments

First of all, I have to say I have read the manuscript from the perspective of a researcher who is interested in inhibitory control but is also extremely critical of how some people (over)use the concept of inhibition, which has become meaningless. And I am afraid the reviewed manuscript highlights this latter issue (even when I am not putting the blame on the authors. I understand they are using tasks and measures that many others think are ‘inhibitory’ in nature).

The manuscript is very well written and clear, and I do appreciate its transparency. The main goal of the study is theoretically relevant and timely and the statistical approach is appropriate and well reported.

That said, I have also some (theoretically-grounded) concerns with the manuscript that essentially stem from how the concept ‘inhibition’ is (mis)used (not necessarily by the authors of the manuscript but it is clear they rely on it) and mainly focus on the conclusions to be drawn from the results of the study. Actually, some of my concerns are already mentioned in the Discussion of the paper (but poorly addressed).

First, the selection of the tasks noticeably did bias the latent variables that were (or not) identified. It seems obvious that the structural similarities shared by the three updating tasks maximized the high correlations between their measures. The opposite seems to be the case regarding the inhibition tasks, which only have one thing in common; the need of conflict/interference resolution in some experimental conditions. However, the conflict source, the target of control, the level of processing wherein conflict arose and the mechanisms (maybe inhibition, maybe not) to be triggered to deal with the different types of conflict substantially differed across tasks. Moreover, none of tasks directly relate to managing information in working memory. Hence, even in the case inhibition was the construct underlying the six conflict resolution tasks, why should we expect performance on them correlate with performance on updating/removal tasks?

I know the answer: some researchers have claimed that inhibition (as an omnipresent factor) is also involved in managing contents in working memory.

My thought is that the best way to test this idea is to define/precise how inhibition could act on irrelevant/outdated information in WM and, if possible, to build up experimental procedures to disentangle inhibition from any other potential mechanism*. Otherwise, removal, updating, inhibition or any other concept will remain blurred and theoretically vague. Moreover, I do not think methodological approaches like the one used in the reviewed manuscript helps because they rely on imprecise constructs and hard to interpret findings. In fact, in my opinion the ‘main’ contribution of the reviewed manuscript is to reveal that the latent variable underlying three very similar updating tasks is not the same as the construct(s) that is(are) measured by a set of very different conflict-resolution tasks. The study says nothing about inhibition, because we do not know if different inhibitory mechanisms are involved in the used tasks (even in those named ‘removal tasks”). If the mechanism is not well defined in computational terms to describe what is inhibited and how, and it is not well supported by behavioral and brain-related measures to illustrate the inhibitory process and its consequences, in reality what we call inhibition could be anything.

As a final point, I was shocked by the rationale for not using the Stop-Signal Task as an inhibitory task. I do not know a task that is totally free of strategical thinking (do you?). Despite this, versions of this task exist that minimizes the use of strategies. The SST is widely considered a “process pure” task to measure inhibitory control (very precisely defined, by the way). Anyway, personally I wouldn’t expect the SSRT (the inhibitory index from the SST) to correlate with removal as a construct (despite it could be the case that the removal process could have an inhibitory component as well).

*Examples of this now exist in the literature on (episodic memory) inhibition where behavioral and brain-related measures can straightforwardly be interpreted from an inhibitory view after ruling out alternative non-inhibitory mechanisms. See for example:

- Weller et al. (2013). On the status of cue independence as a criterion for memory inhibition: evidence against the covert blocking hypothesis. J. Exp. Psychol. Learn. Mem. Cognit. 39 (4), 1232–1245.

- Wimber et al. (2015). Retrieval induces adaptive forgetting of competing memories via cortical pattern suppression. Nat. Neurosci. 18 (4), 582–589.

Reviewer #2: The authors report an experimental study aimed to test whether two distinct processes strongly related to working memory (i.e., item removal/updating and inhibition) are the same construct or not. To this purpose, the authors 1) administered various tasks assumed to tap into the two processes under investigation, and 2) adopted a confirmatory factor analysis approach to explore the possible correlational structure of the different measures. The results are interpreted as tentative evidence that removal and inhibition are independent processes.

This is a study addressing a timely issue, whose results might be interesting for a wide community of cognitive scientists interested in working memory and executive functioning. The study looks well conducted overall, predictions are clear, the analyses seem elegant and informative, and writing is very clear. There are only a few points that, in my view, should be addressed in a revision.

1. Methods section. More details should be provided concerning the number of trials (and potential data points) for each task.

2. I think that the discussion section would benefit from including the analysis of a recent paper by Draheim et al. (2020) which is related to the issues addressed in the present paper. I believe the authors should acknowledge the potential problems arising from addressing individual differences investigations by means of RT tasks.

Here is the reference: Draheim, C., Tsukahara, J. S., Martin, J. D., Mashburn, C. A., & Engle, R. W. (2020). A toolbox approach to improving the measurement of attention control. Journal of Experimental Psychology: General. In press. https://doi.org/10.1037/xge0000783

3. In the general discussion, I think it would be fair to cite other tasks that are often used to investigate inhibitory processing. These include tasks belonging to the go/no-go family (e.g., the Sustained Attention to Response Task), but also more complex paradigms addressing episodic memory such as the Retrieval Practice Paradigm.

6. PLOS authors have the option to publish the peer review history of their article (what does this mean?). If published, this will include your full peer review and any attached files.

Reviewer #1: No

Reviewer #2: No

---

## [Author Response · Author response to Decision Letter 0]

4 Aug 2020

Reviewer A

A1 “First, the selection of the tasks noticeably did bias the latent variables that were (or not) identified. It seems obvious that the structural similarities shared by the three updating tasks maximized the high correlations between their measures. The opposite seems to be the case regarding the inhibition tasks, which only have one thing in common; the need of con-flict/interference resolution in some experimental conditions. However, the conflict source, the target of control, the level of processing wherein conflict arose and the mechanisms (maybe inhibition, maybe not) to be triggered to deal with the different types of conflict sub-stantially differed across tasks. Moreover, none of tasks directly relate to managing infor-mation in working memory. Hence, even in the case inhibition was the construct underlying the six conflict resolution tasks, why should we expect performance on them correlate with performance on updating/removal tasks?

I know the answer: some researchers have claimed that inhibition (as an omnipresent factor) is also involved in managing contents in working memory.

My thought is that the best way to test this idea is to define/precise how inhibition could act on irrelevant/outdated information in WM and, if possible, to build up experimental procedures to disentangle inhibition from any other potential mechanism*. Otherwise, removal, updating, inhibition or any other concept will remain blurred and theoretically vague. Moreover, I do not think methodological approaches like the one used in the reviewed manuscript helps because they rely on imprecise constructs and hard to interpret findings. In fact, in my opinion the ‘main’ contribution of the reviewed manuscript is to reveal that the latent variable underlying three very similar updating tasks is not the same as the construct(s) that is(are) measured by a set of very different conflict-resolution tasks. The study says nothing about inhibition, be-cause we do not know if different inhibitory mechanisms are involved in the used tasks (even in those named ‘removal tasks”). If the mechanism is not well defined in computational terms to describe what is inhibited and how, and it is not well supported by behavioral and brain-related measures to illustrate the inhibitory process and its consequences, in reality what we call inhibition could be anything. 

*Examples of this now exist in the literature on (episodic memory) inhibition where behavioral and brain-related measures can straightforwardly be interpreted from an inhibitory view after ruling out alternative non-inhibitory mechanisms. See for example:

- Weller et al. (2013). On the status of cue independence as a criterion for memory inhibition: evidence against the covert blocking hypothesis. J. Exp. Psychol. Learn. Mem. Cognit. 39 (4), 1232–1245.

- Wimber et al. (2015). Retrieval induces adaptive forgetting of competing memories via corti-cal pattern suppression. Nat. Neurosci. 18 (4), 582–589.”

RESPONSE: In order to take Reviewer A’s point of view into account, we have clarified more explicitly in the Discussion section (see p. 34) that our main conclusion with regards to inhibi-tion is indeed that performance on the inhibition tasks we used relies on interference-resolution processes that are different from removal. We also agree with the reviewer that this provides additional evidence that we should stop calling everything inhibition, and that more work needs to be done to tease apart specific conflict-resolution processes. 

The new section reads: "Together, the present results show that performance on the inhibition tasks we used relies on interference-resolution processes that are different from removal. On a general level, the results also provide additional evidence that we should stop using inhibition as an umbrella term for a variety of distinct processes. Thus, although the present study sup-ports the notion that removal is not an inhibitory process, in order to progress, the field will need to specify how inhibition could act on irrelevant/outdated information in WM, and should attempt to build experimental paradigms that are able to disentangle inhibition from other mechanisms of conflict resolution (as has been done with inhibition in episodic memory [Weller et al., 2013; Wimber et al., 2015]).” 

A2 “As a final point, I was shocked by the rationale for not using the Stop-Signal Task as an inhibitory task. I do not know a task that is totally free of strategical thinking (do you?). De-spite this, versions of this task exist that minimizes the use of strategies. The SST is widely considered a “process pure” task to measure inhibitory control (very precisely defined, by the way). Anyway, personally I wouldn’t expect the SSRT (the inhibitory index from the SST) to correlate with removal as a construct (despite it could be the case that the removal process could have an inhibitory component as well).”

RESPONSE: We now address this in the Discussion, clarifying why we did not include the stop-signal task (p.35-36): “Although there exist versions of the stop-signal task that minimize the use of strategies (e.g., by implementing monetary incentives [see Leotti & Wager, 2010]), these versions have rarely been used in individual-differences research (Chuderski et al., 2012; Friedman & Miyake, 2004; Klauer et al., 2010; Miyake et al., 2000; Rey-Mermet et al., 2018, but see 2019 for one exception). As stated in the introduction, we opted for inhibition tasks that are broadly assumed to require inhibition and have been commonly used in individ-ual-differences research. Similarly, we did not include the SART because the dependent measures typically used in this task (e.g., accuracy rates on no-go trials or the signal detection measure d’) do not control for variance in processing speed. If fast and slow individuals re-spond within about the same time, slower individuals will make more errors, thus transferring some speed variance onto accuracy and d’.”

Reviewer B

B1 “Methods section. More details should be provided concerning the number of trials (and potential data points) for each task.”

RESPONSE: Details about the number of trials were initially provided in the supplementary Table S1. Nevertheless, we agree with Reviewer B that these details are sufficiently important to be provided in the main text. Therefore, we have now added this table (as the new Table 1) to the main text (p. 10).

B2 “I think that the discussion section would benefit from including the analysis of a recent paper by Draheim et al. (2020) which is related to the issues addressed in the present paper. I believe the authors should acknowledge the potential problems arising from addressing indi-vidual differences investigations by means of RT tasks.”

RESPONSE: We thank the reviewer for pointing this out. We now discuss Draheim et al. (2020) in the Discussion section as follows (see p. 32-33): 

“As pointed out recently by Draheim et al. (2019, 2020), one possible reason for the difficulty in establishing a coherent factor of inhibition might be the use of RTs—and in particular, RT dif-ference scores—because these scores have been associated with low reliability and speed-accuracy trade-offs. Besides the substantial impact of measurement error (Rouder et al., 2019), previous research has emphasized the low performance variability in the tasks used to assess inhibition, resulting in a low test/re-test reliability of these measures (see Cooper et al., 2017; Hedge et al., 2017). However, both the present study and Rey-Mermet et al. (2018) showed that when computed with a sufficient number of trials, RT difference scores can be measured with reasonable internal consistency reliabilities. Perhaps more importantly, in the present study we also disattenuated the correlations for imperfect reliability in the inhibition measures. Moreover, when using a calibration procedure in order to measure inhibition through accuracy, thus pushing speed-accuracy trade-offs into accuracy scores, Rey-Mermet et al. (2019) were still unable to establish a factor of inhibition. Thus, the use of RTs and RT difference scores per se cannot be the sole reason for the difficulties to establish inhibition at the latent-variable level. 

A further reason why the use of difference scores may impede extraction of a latent inhibition variable is that computing difference scores is based on the premise of additive factors (Draheim et al., 2019; Rey-Mermet et al., 2019). However, this premise is questionable be-cause it is not known whether the durations of baseline processes and inhibitory processes combine additively to the RTs of incongruent trials. It is for this reason that both the present study and Rey-Mermet et al. (2018; 2019) applied a bi-factor modeling approach to overcome the premise of additive factors. In this approach, all incongruent and congruent trials were forced to load on a baseline factor and the incongruent trials were forced to load on an inhibi-tion factor. However, even when using a bifactor model, no inhibition-specific variance could be established. Thus, the use of difference scores—either in RTs or accuracy rates—can also not totally explain the difficulty of establishing an inhibition factor at the latent-variable level.”

B3 “In the general discussion, I think it would be fair to cite other tasks that are often used to investigate inhibitory processing. These include tasks belonging to the go/no-go family (e.g., the Sustained Attention to Response Task), but also more complex paradigms addressing epi-sodic memory such as the Retrieval Practice Paradigm.” 

RESPONSE: The Discussion now mentions the go/no-go task (or Sustained Attention to Re-sponse Task) and the reason it was not included in the present study (see p. 36): 

“Similarly, we did not include the SART because the dependent measures typically used in this task (e.g., accuracy rates on no-go trials or the signal detection measure d’) do not control for variance in processing speed. If fast and slow individuals respond within about the same time, slower individuals will make more errors, thus transferring some speed variance onto accuracy and d’.”

We decided not to mention the retrieval practice paradigm because this task does not belong to the tasks commonly used to assess inhibition in individual-differences research focusing on attentional control/executive functions.

---

## [Decision Letter · Decision Letter 1]

23 Sep 2020

PONE-D-20-13452R1

Removal of information from working memory is not related to inhibition

PLOS ONE

Dear Dr. Alodie Rey-Mermet

Thank you for submitting your manuscript to PLOS ONE. After careful consideration, we feel that it has merit but does not yet meet PLOS ONE’s publication criteria as it currently stands. Therefore, we invite you to submit a revised version of the manuscript that addresses the points raised during the review process.

ACADEMIC EDITOR: Dear Dr. Alodie Rey-Mermet,

as you will see, the reviews are mixed. In particular, one of the Reviewer is satisfied with the changes you have made. The other one still see some substantial -theoretical-  weakness in your study. I have  also looked at the revised version of your interesting paper. I have to say that I agree with the comments done by Reviewer 1 . I am thus asking you to revise the paper carefully taking into consideration the points made by Reviewer 1 that mainly regards the introduction and the discussion section. The points the Reviewer raised are important from a theoretical point of view,  and by considering  and "including" them in your revision (which means also clarifying some debated aspects on the construct you are working in), your paper will have a strong impact in the literature on inhibition. 

We look forward to receiving your revised manuscript.

Kind regards,

Erika Borella, Ph.D.

Academic Editor

PLOS ONE

Reviewers' comments:

Reviewer's Responses to Questions

**Comments to the Author**

1. If the authors have adequately addressed your comments raised in a previous round of review and you feel that this manuscript is now acceptable for publication, you may indicate that here to bypass the “Comments to the Author” section, enter your conflict of interest statement in the “Confidential to Editor” section, and submit your "Accept" recommendation.

Reviewer #1: (No Response)

Reviewer #2: All comments have been addressed

2. Is the manuscript technically sound, and do the data support the conclusions?

Reviewer #1: Partly

Reviewer #2: Yes

3. Has the statistical analysis been performed appropriately and rigorously? 

Reviewer #1: Yes

Reviewer #2: Yes

4. Have the authors made all data underlying the findings in their manuscript fully available?

Reviewer #1: Yes

Reviewer #2: Yes

5. Is the manuscript presented in an intelligible fashion and written in standard English?

Reviewer #1: Yes

Reviewer #2: Yes

6. Review Comments to the Author

Reviewer #1: These are my thoughts after reading the revised manuscript carefully.

To be honest, I thought my comments to the previous version of the manuscript would prompt the authors to somehow refocus the introduction/discussion (they seem to agree with me in relevant theoretical points according to their responses on the letter to reviewers). This has not been the case, however, with the revised manuscript only including a new few lines in the discussion. Well, this is their manuscript, not mine. However, I do not think the manuscript has been significantly improved.

If the authors (like me) believe that their ‘inhibition’ tasks mostly tap into (mechanism-free) interference control (and the discussion now seems to endorse this idea), the introduction should also bring this into focus to increase coherence and consistency throughout the text. Potential readers would benefit from it.

By the way, the new paragraph in the discussion includes the following statement: ‘Thus, although the present study supports the notion that removal is not an inhibitory process…’. Again, I DO NOT think this is true. In fact, it seems quite contradictory to the idea that maybe the used ‘inhibition’ tasks do not measure inhibition.

Related to this, it is now clear to me that the title of the manuscript should be changed. I cannot see the point of having “inhibition” in it.

To conclude, a couple of notes aimed to encourage the authors to rethink some of the arguments they deploy on the manuscript/response letter:

1. The Stop-Signal Task has been widely used to investigate individual differences (a recent example here https://www.jneurosci.org/content/38/36/7887). Thus, to justify that you did not pick this task for your study, you better use a different reason.

2. In response to a (very smart) suggestion by the other reviewer, the authors write: ‘We decided not to mention the retrieval practice paradigm because this task does not

belong to the tasks commonly used to assess inhibition in individual-differences’. Well, the authors could want to know that the manuscript already includes mentions to the retrieval practice paradigm (‘… the field will need to specify how inhibition could act on irrelevant/outdated information in WM, and should attempt to build experimental paradigms that are able to disentangle inhibition from other mechanisms of conflict resolution (as has been done with inhibition in episodic memory [73,74]’). 73 and 74 refer to retrieval practice studies. Also, and despite the authors think, the RP (sometimes named ‘retrieval-induced forgetting’) paradigm has been extensively used to explore individual differences in inhibitory control [over episodic memory, of course; have a look at the meta-analytic review by Murayama et al. (2014) here https://pubmed.ncbi.nlm.nih.gov/25180807/].

Obviously, I totally endorse the original suggestion made by the other reviewer.

Reviewer #2: (No Response)

7. PLOS authors have the option to publish the peer review history of their article (what does this mean?). If published, this will include your full peer review and any attached files.

Reviewer #1: No

Reviewer #2: No

---

## [Author Response · Author response to Decision Letter 1]

3 Nov 2020

Editor 

E1 “As you will see, the reviews are mixed. In particular, one of the Reviewer is satisfied with the changes you have made. The other one still see some substantial -theoretical- weakness in your study. I have also looked at the revised version of your interesting paper. I have to say that I agree with the comments done by Reviewer 1 . I am thus asking you to revise the paper carefully taking into consideration the points made by Reviewer 1 that mainly regards the introduction and the discussion section. The points the Reviewer raised are important from a theoretical point of view, and by considering and "including" them in your revision (which means also clarifying some debated aspects on the construct you are working in), your paper will have a strong impact in the literature on inhibition.”

RESPONSE: We revised the manuscript carefully, in particular the introduction and the discussion, and we clarified the formulation regarding the concept of inhibition where appropriate (see p. 2-3, 5-7, 20, 24, 28, 30-31, 33-34, and 37-38). Nevertheless, we still use the descriptive terms “inhibition task” or “inhibition factor” in the manuscript. Given their use is convention, we think that this allows readers to readily grasp what task or factor we are referring to.

Reviewer A

A1 “To be honest, I thought my comments to the previous version of the manuscript would prompt the authors to somehow refocus the introduction/discussion (they seem to agree with me in relevant theoretical points according to their responses on the letter to reviewers). This has not been the case, however, with the revised manuscript only including a new few lines in the discussion. Well, this is their manuscript, not mine. However, I do not think the manuscript has been significantly improved.

If the authors (like me) believe that their ‘inhibition’ tasks mostly tap into (mechanism-free) interference control (and the discussion now seems to endorse this idea), the introduction should also bring this into focus to increase coherence and consistency throughout the text. Potential readers would benefit from it.”

RESPONSE: We see the Reviewer’s point. However, we were reluctant to reframe the introduction too much in order to maintain an accurate representation of our motivation. However, we also agree that coherence and consistency are important. Thus, as suggested by Reviewer A, we now specify in the introduction that the inhibition tasks mostly tap into task-specific interference-control processes (see p. 5-6): “Overall, this research has led some authors to the conclusion that the tasks used to assess inhibition only assess the ability to reduce the interference arising in that particular task (Rey-Mermet et al., 2018, 2019). In this case, the inhibition tasks mostly tap into interference-control processes, which are highly task specific.

In summary, previous individual-differences research has put forward inhibition as a core interference-control mechanism related to WM. Nevertheless, there is some evidence that establishing inhibition as a valid and reliable construct at the latent-variable level is more difficult than previously thought. This emphasizes that inhibition might not be a general construct, thus mandating us to be very cautious when referring to the concept of “inhibition”.”

Please note that we opted to use the term “task-specific” instead of “mechanism-free”. In our opinion, the term “mechanism-free” might lead readers to conclude that the inhibition tasks involve no interference-control processes at all. However, in order for participants to perform the tasks, we think that some interference-control processes are required. Whether these processes include an active suppression of irrelevant information, a strengthened activation of the correct response, and/or simply the passive decay of the interference from the irrelevant information is still a question of debate and an issue for future research. However, Rey-Mermet et al. (2018, 2019) as well as the present results clearly show that these processes did not generalize across the different inhibition tasks, explaining why we preferred the terminology of “task-specific” interference-control processes. 

A2 “By the way, the new paragraph in the discussion includes the following statement: ‘Thus, although the present study supports the notion that removal is not an inhibitory process…’. Again, I DO NOT think this is true. In fact, it seems quite contradictory to the idea that maybe the used ‘inhibition’ tasks do not measure inhibition.”

RESPONSE: We changed the sentence as follows (see p. 34): “although the present study supports the notion that removal is not related to the interference-control processes assessed by the inhibition tasks”.

A3 “Related to this, it is now clear to me that the title of the manuscript should be changed. I cannot see the point of having “inhibition” in it.”

RESPONSE: We changed the title as follows (see p. 1): “Interference control in working memory: Evidence for discriminant validity between removal and inhibition tasks”. We think that having “inhibition” in the title is important given the paper might be of interest to scholars interested in inhibition. 

A4 “The Stop-Signal Task has been widely used to investigate individual differences (a recent example here https://www.jneurosci.org/content/38/36/7887). Thus, to justify that you did not pick this task for your study, you better use a different reason.”

RESPONSE: We thank Reviewer A for pointing out this imprecision. Our point was that the versions of the stop-signal task that minimize the use of strategies have rarely been used in individual-differences research using a CFA and SEM approach, such as the approach used in the present study. This has now been clarified in the manuscript (see p. 36).

A5 “In response to a (very smart) suggestion by the other reviewer, the authors write: ‘We decided not to mention the retrieval practice paradigm because this task does not belong to the tasks commonly used to assess inhibition in individual-differences’. Well, the authors could want to know that the manuscript already includes mentions to the retrieval practice paradigm (‘… the field will need to specify how inhibition could act on irrelevant/outdated information in WM, and should attempt to build experimental paradigms that are able to disentangle inhibition from other mechanisms of conflict resolution (as has been done with inhibition in episodic memory [73,74]’). 73 and 74 refer to retrieval practice studies. Also, and despite the authors think, the RP (sometimes named ‘retrieval-induced forgetting’) paradigm has been extensively used to explore individual differences in inhibitory control [over episodic memory, of course; have a look at the meta-analytic review by Murayama et al. (2014) here https://pubmed.ncbi.nlm.nih.gov/25180807/]. Obviously, I totally endorse the original suggestion made by the other reviewer.”

RESPONSE: We again thank Reviewer A for pointing out another imprecision. Previous research has clearly shown that inhibition in episodic memory – as assessed with the retrieval practice paradigm – differs from the interference-control processes measured in tasks such as the Stroop and flanker tasks (e.g., Friedman & Miyake, 2004; Pettigrew & Martin, 2014). Based on this research, we made a distinction between the interference-control processes measured with paradigms such as the retrieval practice paradigm (which we refer to as “inhibition in episodic memory”) and the interference-control processes measured with the inhibition tasks, such as those used in the present study. Therefore, when we wrote in our previous response that the retrieval practice paradigm “does not belong to the tasks commonly used to assess inhibition in individual-differences research”, we meant that this paradigm does not belong to the tasks typically called “inhibition tasks” (e.g., Stroop flanker, Simon, antisaccade tasks).

---

## [Decision Letter · Decision Letter 2]

16 Nov 2020

Interference control in working memory: Evidence for discriminant validity between removal and inhibition tasks

PONE-D-20-13452R2

Dear Dr. Alodie Rey-Mermet,

We’re pleased to inform you that your manuscript has been judged scientifically suitable for publication and will be formally accepted for publication once it meets all outstanding technical requirements.

Kind regards,

Erika Borella, Ph.D.

Academic Editor

PLOS ONE

Additional Editor Comments (optional):

I am pleased to accept this interesting paper.

Reviewers' comments:

Reviewer's Responses to Questions

**Comments to the Author**

1. If the authors have adequately addressed your comments raised in a previous round of review and you feel that this manuscript is now acceptable for publication, you may indicate that here to bypass the “Comments to the Author” section, enter your conflict of interest statement in the “Confidential to Editor” section, and submit your "Accept" recommendation.

Reviewer #1: All comments have been addressed

2. Is the manuscript technically sound, and do the data support the conclusions?

Reviewer #1: Yes

3. Has the statistical analysis been performed appropriately and rigorously? 

Reviewer #1: Yes

4. Have the authors made all data underlying the findings in their manuscript fully available?

Reviewer #1: Yes

5. Is the manuscript presented in an intelligible fashion and written in standard English?

Reviewer #1: Yes

6. Review Comments to the Author

Reviewer #1: (No Response)

7. PLOS authors have the option to publish the peer review history of their article (what does this mean?). If published, this will include your full peer review and any attached files.

Reviewer #1: No

---

## [Editor Report · Acceptance letter]

20 Nov 2020

PONE-D-20-13452R2 

Interference control in working memory:
Evidence for discriminant validity between removal and inhibition tasks 

Dear Dr. Rey-Mermet:

I'm pleased to inform you that your manuscript has been deemed suitable for publication in PLOS ONE. Congratulations! Your manuscript is now with our production department. 

Kind regards, 

on behalf of

Dr. Erika Borella 

Academic Editor

PLOS ONE